# SELECTIVE ATTENTION IMPROVES TRANSFORMER

**Yaniv Leviathan**
Google Research
leviathan@google.com

**Matan Kalman**
Google Research
matank@google.com

**Yossi Matias**
Google Research
yossi@google.com

## ABSTRACT

Unneeded elements in the attention's context degrade performance. We introduce *Selective Attention*, a simple parameter-free change to the standard attention mechanism which reduces attention to unneeded elements. Selective attention consistently improves language modeling and downstream task performance in a variety of model sizes and context lengths. For example, transformers trained with the language modeling objective on C4 with selective attention perform language modeling equivalently to standard transformers with ∼**2X** more heads and parameters in their attention modules. Selective attention also allows decreasing the size of the attention's context buffer, leading to meaningful reductions in the memory and compute requirements during inference. For example, transformers trained on C4 with context sizes of 512, 1,024, and 2,048 need **16X**, **25X**, and **47X** less memory for their attention module, respectively, when equipped with selective attention, as those without selective attention, with the same validation perplexity.

## 1 INTRODUCTION

Different tasks have different memory requirements. On one extreme, copying an arbitrary sequence requires retaining all sequence elements in memory. On the other extreme, determining whether a specific element appeared at least once, only requires persisting a constant amount of memory.

Transformers (Vaswani et al., 2017) keep the entire history in their context buffers, allowing them to solve tasks such as copying, while famously leading to their squared attention cost. RNNs (Rumelhart et al., 1986) and their modern structured state space variants (Gu et al., 2022; Gu & Dao, 2024) keep only a constant-sized sketch of the history, making inference cost linear, but rendering them incapable of solving tasks such as arbitrary string copying.

*Can we design a model that persists just the right amount of context?*

Several works (see Section 8) aim to improve costs by compressing or otherwise reducing the context size with minimal impact to quality. We take a different approach, focusing instead on quality improvement, and treating cost reductions as a side benefit. Specifically, it has been demonstrated (Leviathan, 2022) that for some tasks removing unneeded elements from the context buffer enables more efficient transformer programs. Indeed, in the attention's differentiable memory, all memory cells contribute to the data read, and circuitry is needed to filter out the noise generated by irrelevant memories. Reducing the amount of circuitry needed should improve performance.

In this work we propose *Selective Attention*, a simple extension to the standard attention mechanism which allows a token to decide that *another* token is no longer needed, reducing the attention that future tokens will pay to it. Selective attention adds no new parameters and only a negligible amount of computation, yet yields meaningful improvements in synthetic tasks and natural language modeling for a range of model and context sizes. Additionally, we show that elements that are selected to be forgotten by selective attention can be safely removed from the attention's context, leading to substantial reductions in the memory and computation requirements during inference, without penalizing quality. We name our method after the related neuroscience concept of selective attention. Quoting Plebanek & Sloutsky (2017): "Selective attention allows adults to focus on task-relevant information, while ignoring task-irrelevant information. This in turn leads to superior processing of task-relevant information."

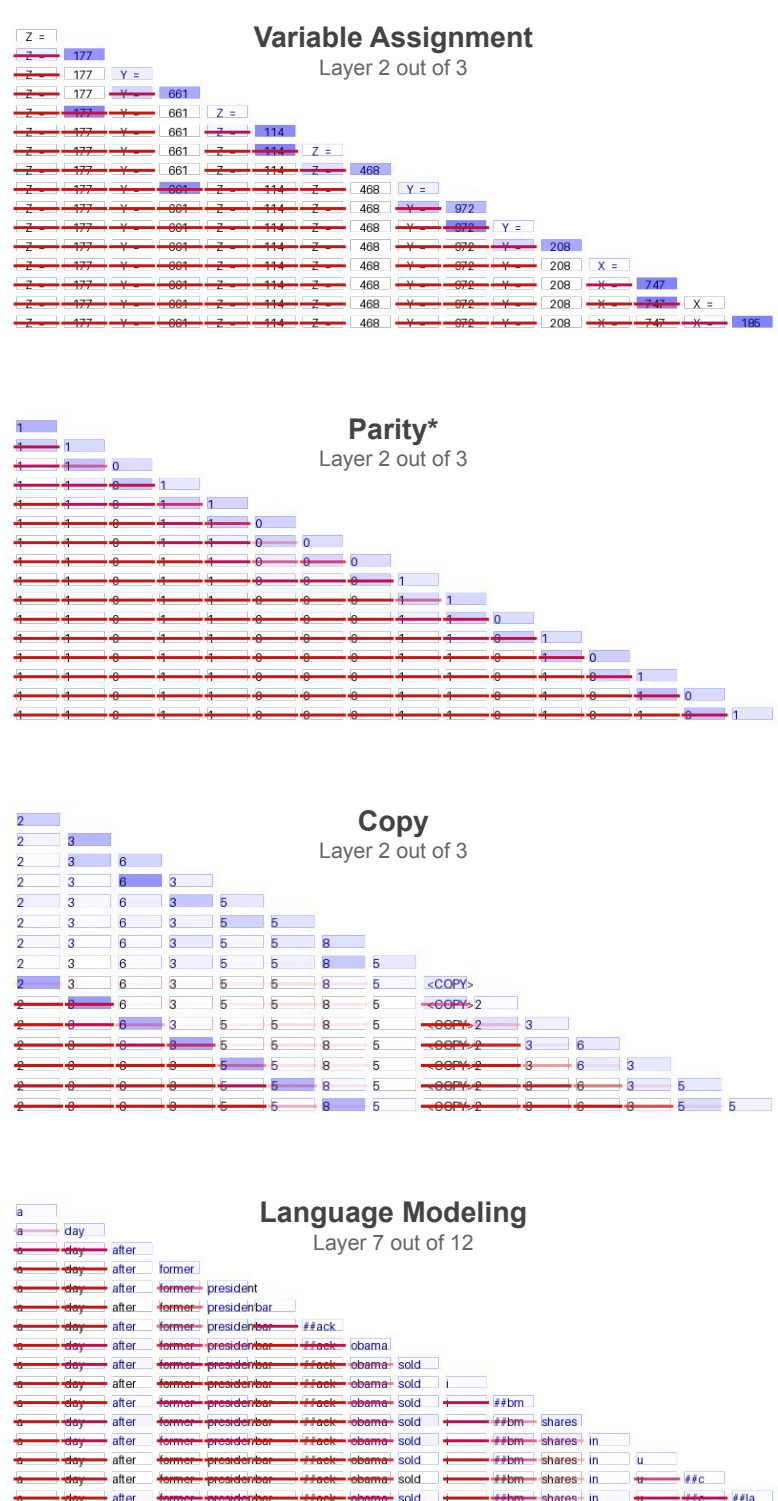

Figure 1: A visualization of the masking by selective attention (red strike-through) and attention strength (averaged across heads, blue highlight) for different tasks (see Section 2).

## 2    MOTIVATING EXAMPLES

Consider a transformer processing an input sequence with three tokens: `a`, `b`, `c`. In a given layer with the standard attention module, token `b` can decide how much to read from token `a`, and token `c` can decide how much to read from token `a`, but *token `b` cannot affect how much token `c` reads from token `a`*. Specifically, if token `b` has determined that token `a` is irrelevant or even misleading to future tokens such as `c`, there is nothing it can do in the given layer to correct for this. Even in subsequent layers, masking token `a` is not trivial. Selective attention enables exactly such masking. To illustrate its usefulness, let's consider the `Variable Assignment` problem, as well as natural language modeling.

In `Variable Assignment` the input consists of a set of repeated assignments to named variables, followed by a query for the latest value for one of the variables which the model needs to output. For example, for the input: `y=7; x=1; x=3; z=5; x=?` the output is `3`. Note that `Variable Assignment` can be seen as a generalization of the `Search` problem (Leviathan, 2022), where repeated occurrences of the query pattern are allowed, and we are tasked with finding the most recent occurrence. It is well known that the `Search` problem is easily solvable by standard transformers, via induction heads (Olsson et al., 2022). Selective attention facilitates a simple reduction from `Variable Assignment` to `Search`, whereby every assignment to a variable masks out all previous assignments to the same variable. In Figure 1 (top) we see that this is indeed the case for a transformer trained with selective attention. In Appendix A.1 we show that a transformer with selective attention easily learns a general solution to `Variable Assignment` while a standard transformer does not.

To motivate selective attention for natural language modeling, we first note that `Variable Assignment` is a common sub-task, e.g. when persisting a state. For further motivation, consider the common case where a part of the input is ambiguous, and the ambiguity is only resolved at a later token. For example, in the sequence: `Bar, ##ack, Obama`, the first token `Bar` encodes several competing meanings, but the later tokens `##ack` and `Obama` resolve it to the entity for the ex-president. For many tasks that are mostly concerned with the semantic meaning, later tokens might not want to read the ambiguous meaning from the earlier tokens, so masking them, as with selective attention, might be useful. In Figure 1 (bottom) we see that this is indeed the case for a transformer trained with selective attention. In the visualized layer, the last token in multi-token expressions masks out the earlier tokens. For example, `##ack` masks out `bar`; `obama` masks out both `bar` and `##ack`; `##bm` masks out `i`; and `##la` masks out both `u` and `##c`. We also observe additional masking, e.g. the token `after` masks out the tokens `a` and `day`, perhaps because the token `after` absorbed the meaning from the tokens `a` and `day`, or perhaps because the model deems the extra detail is not helpful at this point.

Figure 1 also shows that for the trivial task of `Parity`*, where intermediate results are stored every other token, so that the model's output is only a function of the last two tokens, everything but the last two tokens is masked. For the `Copy` task, selective attention persists the entirety of the string to be copied until copying starts, and then masks out every element as it is copied. See Appendix A.2.

## 3    SELECTIVE ATTENTION

Selective attention is a simple modification on top of standard attention. The key idea is that tokens can *mask* previous tokens, i.e. the amount of attention a token `c` pays a previous token `a` can be reduced by the tokens located between `a` and `c`. For context size $N$, we produce a real-valued $N \times N$ masking matrix $S$ where $S_{i,j}$ denotes how much token $x_i$ masks token $x_j$ (see Section 3.1). We then constrain $S$, e.g. to be causal and non-negative (see Section 3.2). We finally accumulate the information in matrix $S$ into a new matrix $F$, taking into account masking by *all* previous tokens (see Section 3.3). The matrix $F = \text{Accumulate}(\text{Constrain}(S))$ is then simply subtracted from the attention logits before applying the softmax:

$$\text{SelectiveAttention}(Q, K, V) = \text{softmax}(\frac{QK^T}{\sqrt{d_k}} - F)V \tag{1}$$

Figure 2 illustrates a sketch implementation.

## 3.1 SELECTION FUNCTION

Computing the selection matrix $S$ is akin to computing a compatibility score, and many functions can be used, for example, a separate bilinear form. Following an observation from Leviathan (2022) that a common case for a token wanting to mask another is after *absorbing* its contents (i.e. after attending to it), we instead simply reuse the result of one of the existing heads of the attention module. This means that the selection function adds no new parameters or computation. Note that the head still contributes to attention as usual. We also experimented with a separate bilinear form, and in spite of adding additional parameters and computation, this resulted in the same or slightly worse results (see Appendix A.3).

## 3.2 CONSTRAINTS

Following observations from Leviathan (2022), we apply the following constraints to $S$:

1. Zero out negative values (i.e. applying `ReLU`), only reducing attention, never boosting it.
2. Zero out the first column, so as not to mask the `<BOS>` token.
3. Zero out the diagonal, so as not to let a token mask itself.

We observe that all three constraints improve performance (see Appendix A.5).

## 3.3 ACCUMULATION

In this work we focus on transformer decoders, so selective attention cannot influence the attention operation by past tokens. We chose cumulative summation as our accumulation function. We observe some improvement by only applying the masking for *future* tokens (i.e. the masking by a token would not affect its own attention operation), so $F_{i,j} = \sum_{k \leq i-1} S_{k,j}$ (see ablation in Appendix A.4).

```
...
attn_logits = einsum("bhnd,bhmd->bhnm", Q, K) / sqrt(dk)
attn_logits = where(causal_mask, attn_logits, float("-inf"))
S = attn_logits[:, 0]                # Select head 0.
S = relu(S)                          # Only positive selection.
S[..., 0] = 0                        # Do not mask <BOS>.
S = (1 - eye(n)) * S                 # Do not mask self.
S = roll(S, 1, -2); S[..., 0, :] = 0 # Mask strictly in the future.
F = np.cumsum(S, axis=-2)            # Accumulate.
attn_logits -= F[:, None]
attn_weights = softmax(attn_logits)
...
```

Figure 2: A sketch implementation of selective attention. The colored lines are the additions to standard attention.

# 4 CONTEXT PRUNING

As presented in Section 3, while beneficial to model quality, selective attention has negligible impact on inference efficiency[1]. However, an additional modification can improve inference efficiency substantially. Specifically, selective attention can reduce the memory and computation requirements of the attention module, via pruning elements from the context buffer.

To see how, note that once a token is sufficiently masked by selective attention, it will not contribute meaningfully *to any future attention operations*. Such tokens can be safely evicted from the context buffer. We could pick a fixed threshold, and prune all elements whose soft masking is higher than the threshold (i.e. $F_{i,j} > \tau$), but that would make the memory savings hard to take advantage of (e.g.

---

[1]The additional computation of $\mathcal{O}(hn^2)$ is negligible compared to the $\mathcal{O}(dn^2)$ of standard attention.

due to fragmentation and a variable number of dropped tokens each iteration). Instead, we observe that the sparsity (i.e. the magnitude of the masking) per layer is stable across samples (see Section 7). To that effect we set a fixed memory budget for each layer, which directly translates to memory and compute savings. Since when a token is dropped it remains dropped for all future tokens, given a memory budget of $K = K_1, \ldots, K_L$ tokens for each layer, when processing the first $K_l$ tokens in layer $l$ we drop nothing, and for each following token we drop the single not-yet-dropped past token with the highest $F$ value. This maintains no more than $K_l$ tokens in layer $l$. Given an overall memory budget, we allocate it between the layers, using a greedy iterative approach. For context size $N$, we initialize $K_1^0 = N, \ldots, K_L^0 = N$. In each iteration $t$, we set $K_n^t = K_n^{t-1}$ for all $n \neq m$, and $K_m^t = K_m^{t-1} - C$ for a constant $C$ (we use 8 in our experiments), where $m = \text{argmin}_i \mathcal{L}(\cdot | K^{t-1} - C_i)$, where $C_i = (0, \ldots, 0, C, 0, \ldots, 0)$ contains $C$ at the $i$th position. In other words, we iteratively reduce the memory budget of the layer that impacts model performance the least. We stop when model performance reaches a predefined threshold, in our experiments, the performance of a standard transformer without selective attention.

Note that with a low memory budget there might be some discrepancy between training and inference. Fine tuning the model after the budgets have been set (or even better, in each iteration) might be advantageous and lead to larger reductions in memory budgets, but we haven't experimented with this setup yet.

As selective attention's masking is beneficial for the model, we observe significant reductions in context sizes without any auxiliary losses (see Section 6.2). Nevertheless, we can further encourage the model to mask out more elements by adding an explicit term to the loss, like so:

$$\mathcal{L}_{mem} = \mathcal{L} + \epsilon \cdot \frac{\sum_{l=1}^{L} \max_i M_i^l}{L \cdot n_{\neq pad}} \tag{2}$$

Where $\mathcal{L}$ is the standard loss (log-perplexity in the case of language modeling), $\epsilon$ is a small weight factor (in our experiments we set $\epsilon = 0.1$ without further tuning), $L$ is the number of layers, $n_{\neq pad}$ is the number of non-pad tokens, and $M_i^l = i - \sum_{k=1}^{i} \min(F_{i,k}^l, \tau)/\tau$ is our approximation for the memory requirements at the $i$th token for layer $l$ ($0 \leq M_i^l \leq i$). We clamp $F_{i,k}^l$ from above by $\tau$ so as not to reward increasing it indefinitely ($F$ is already clamped from below to 0). We set $\tau = 1$ without further tuning. Since the memory required for a given layer is the maximum memory required for each of the tokens, the loss only considers the maximum among the $M_i^l$s. We observe further reduction in context sizes with this explicit loss term (see Section 6.2).

## 5 EXPERIMENTAL SETUP

In all of our experiments we use a decoder-only transformer with multi-head attention, as in Vaswani et al. (2017), with the following modifications: we use Pre-LN instead of Post-LN (Xiong et al., 2020), learned position encoding, SwiGLU gates instead of MLPs (Shazeer, 2020), normalize the Q and K projections (Dehghani et al., 2023)[2], remove the biases (Raffel et al., 2023), and replace the LayerNorms with RMSNorm (Zhang & Sennrich, 2019). Note that we tested other variants, including a vanilla decoder-only transformer exactly as in (Radford et al., 2019) and observed similar results. We trained our models with the AdamW optimizer with $\beta_1 = 0.9$ and $\beta_2 = 0.999$ for a total of 524,288 steps. We used cosine decay and 1,000 linear warmup steps and a learning rate of 0.005. We repeated some of the experiments with different learning rates and obtained similar results. We used a batch size of 256 and a fixed context size of 512 for all training runs except for the context size experiments (Figure 3 left) where we used a batch of 128. We follow Esser et al. (2024), and parameterize a model size by a parameter $d$ such that $d_{model} = 64d$ and $n_{heads} = n_{layers} = d$ (see Table 8 in Appendix A.13). We trained all of our models on TPUv4s. For the language modeling experiments, we used the C4 (Raffel et al., 2023) dataset with a vocabulary of size 8K tokens built with the SentencePiece tokenizer (Kudo & Richardson, 2018). We repeated some of the experiments with a vocabulary of size 32K and observed similar results. We also ran experiments with WikiText (Merity et al., 2016), and lm1b (Chelba et al., 2014) and observed similar results.

---

[2]For larger models, we observed more cases of divergence when not normalizing the Q and K projections.

## 6 RESULTS

### 6.1 GENERATION QUALITY

Transformers with selective attention perform consistently better, as measured by perplexity on the validation set, across model and context sizes. We also observe consistent improvements on a set of downstream tasks.

Figure 3 (left) compares the validation perplexity for causal language modeling on the C4 dataset of decoder-only $d = 12$ transformer models with and without selective attention, for varying context sizes. Likewise Figure 3 (right) compares validation perplexity with and without selective attention, for varying model sizes with a context length of 512. We observe improvements across model sizes, and that the improvements grow with the size of the context.

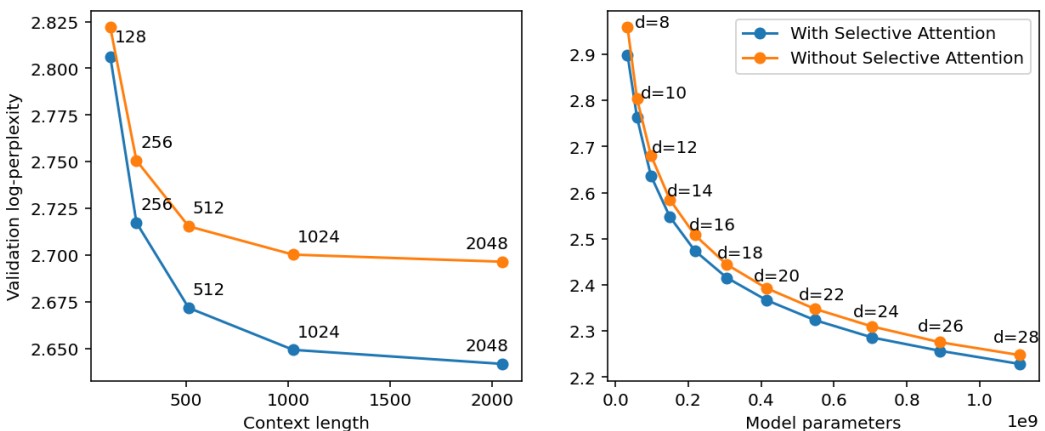

Figure 3: *(Left)* The validation perplexity of a $d = 12$ transformer, with (blue) and without (orange) selective attention, for varying context sizes. *(Right)* The validation perplexity of transformers of various sizes, with (blue) and without (orange) selective attention, for a context size of 512.

Figure 4 shows that even when equipped with additional attention heads (and increasing the parameters of the attention module proportionally, so that the size of each head remains constant), transformers with standard attention only become comparable to those with selective attention, when they have about double the number of heads (and parameters) as their selective attention counterparts.

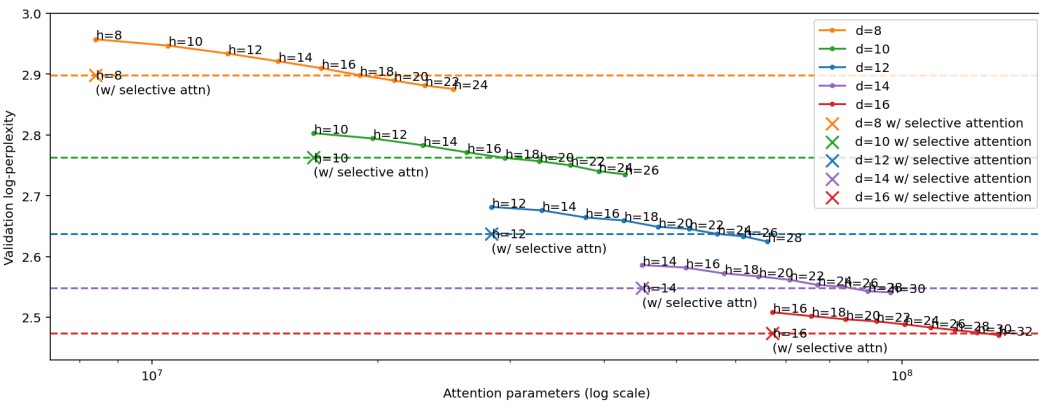

Figure 4: Perplexity of transformers of various sizes with and without selective attention. For the cases without selective attention we add additional attention heads with their respective parameters (i.e. increase the sizes of all projection matrices). Transformers with selective attention perform equivalently to those with standard attention with $\sim$2X as many heads and parameters.

In addition to perplexity on the validation set, we also measure model accuracy on the ARC (Clark et al., 2018), HellaSwag (Zellers et al., 2019), PiQA (Bisk et al., 2019), CommonSenseQA (Talmor et al., 2019), and OpenBookQA (Mihaylov et al., 2018) benchmarks, with and without selective attention, for models of various sizes (see Table 1). We observe consistent improvements across all model sizes with selective attention.

Table 1: Accuracy numbers for baseline transformers with standard attention and transformers with selective attention for various model sizes and downstream tasks.

|  | ARC (Easy) | | ARC (Challenge) | | HellaSwag | |
|---|---|---|---|---|---|---|
|  | Base | Selective | Base | Selective | Base | Selective |
| $d = 16$ | 38.46% | **39.33%** | 23.24% | 23.24% | 38.83% | **40.01%** |
| $d = 18$ | 40.48% | **41.10%** | 24.67% | **24.86%** | 41.77% | **42.60%** |
| $d = 20$ | 41.33% | **41.85%** | 25.14% | **25.52%** | 43.97% | **45.08%** |
| $d = 22$ | 42.43% | **42.70%** | 25.83% | **26.41%** | 46.04% | **47.71%** |
| $d = 24$ | 43.83% | **44.47%** | 26.64% | **26.76%** | 48.70% | **50.32%** |
| $d = 26$ | 44.70% | **45.58%** | 26.37% | **26.95%** | 51.02% | **52.25%** |
| $d = 28$ | 45.64% | **46.95%** | 26.76% | **27.37%** | 53.50% | **53.76%** |

|  | CommonSenseQA | | OpenBookQA | | PiQA | |
|---|---|---|---|---|---|---|
|  | Base | Selective | Base | Selective | Base | Selective |
| $d = 16$ | 25.44% | **25.82%** | 34.26% | **34.33%** | 68.07% | **68.49%** |
| $d = 18$ | 26.45% | **26.69%** | 35.32% | **35.42%** | 69.05% | **69.88%** |
| $d = 20$ | 26.30% | **26.63%** | **35.42%** | 35.27% | 69.82% | **70.49%** |
| $d = 22$ | 26.56% | **27.10%** | 35.96% | **36.88%** | 70.55% | **71.43%** |
| $d = 24$ | 27.49% | 27.49% | 36.36% | **37.13%** | 71.34% | **71.95%** |
| $d = 26$ | 27.70% | **27.78%** | **37.64%** | 37.52% | 71.89% | **72.47%** |
| $d = 28$ | 27.86% | **28.51%** | 37.33% | **37.54%** | 72.32% | **72.85%** |

## 6.2 INFERENCE EFFICIENCY

Efficiency improvements via selective attention stem from a reduction in the attention module's context size when using pruning as in Section 4. Specifically, a smaller context translates directly to more efficient inference in common scenarios. Indeed, note that during inference with a large context and batch size ($bn >> d$), loading the KV-cache (linear in the size of the context) dominates the memory bandwidth (Pope et al., 2022), which is often the bottleneck for generation (Shazeer, 2019). In addition, for very large context sizes ($n >> d$), taking the dot product of the query and the keys in the cache and calculating the weighted average of the values in the cache both dominate compute, i.e., in this setup, a smaller context directly translates to similar gains in FLOPs.

When pruning the context with selective attention, we measure substantial improvements in the memory requirements for the attention module, at the same or better perplexity than the baseline without selective attention. Figure 6 in Appendix A.6 illustrates the trade-off between perplexity and efficiency.

For example, for $d = 12$ transformers with context sizes of 512, 1,024, and 2,048, we see that with selective attention we can maintain the baseline's perplexity while reducing the memory requirements of the attention module by factors of 5X, 7X, and 8X respectively, without any explicit losses (i.e. using only the standard language modeling objective). When training with the $\mathcal{L}_{mem}$ loss (Equation 2) and an $\epsilon$ value of 0.1, the improvements grow to 16X, 25X, and 47X respectively. We also measure the memory savings when considering only very long examples by filtering C4 to only include examples that are at least 90% the size of the context buffer. In that settings we get memory savings of 12X, 18X, and 24X, while maintaining the perplexity of the baseline without selective attention. To maintain the perplexity gains of selective attention, instead of matching the perplexity of the baseline (i.e. the rightmost point on the flat part of the blue graphs of Figure 6), we need 3X, 4X and 4X less memory, for the context sizes above, respectively.

In all cases above, we optimized the per-layer budgets on a training set and reported results on a separate unseen test set.

Finally, we compare selective attention to other attention variants and pruning methods and observe that it achieves a substantially better quality-cost trade-off than all baselines (see Appendix A.10).

# 7 SELECTION PATTERNS

It is interesting to question which elements from the context are being masked by selective attention for language modeling. Figure 5 illustrates the values of the $F$ matrix for a specific example (see Appendix A.7 for the full example text). We observe that some layers (e.g. 6) are mostly dense (i.e. low $F$ values), while other layers (e.g. 2) are sparse (i.e. high $F$ values). As expected, all layers persist some of the most recent elements, but several of the sparse layers (e.g. layers 1, 4, and 9) also persist elements for long time periods, as can be seen by the vertical lines. This suggests that simply limiting the attention module to a local window would not result in the same quality gains as those achieved by selective attention, which we confirm in Appendix A.8.

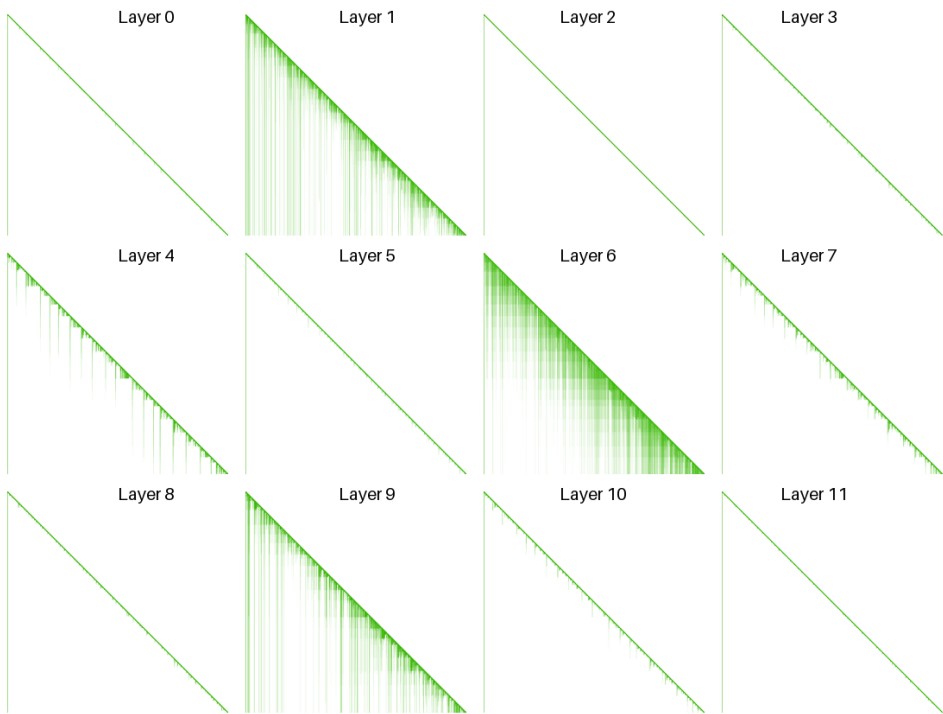

Figure 5: Visualization of the $F$ matrix (greener is lower, i.e. less masking) for a $d = 12$ transformer for the text in Appendix A.7.

Figure 9 in Appendix A.12 illustrates the values of the $F$ matrix averaged over 1,000 examples, and demonstrates that the sparsity patterns are stable across examples. Interestingly, we observe that these sparsity patterns are sometimes stable across different training runs, hinting at some general properties of language modeling (see Figure 10 in Appendix A.12).

Figure 11 in Appendix A.12 depicts the items remaining in the context buffer after pruning (see Section 4), with a budget set to match the perplexity of a standard transformer. We observe that the per-layer budgets correspond well to the values of the $F$ matrix as seen in Figures 5 and 9. For example, layer 6 which gets the highest budget also has the lowest $F$ values. This might indicate that the scales of the $F$ values are consistent across layers.

Figure 12 in Appendix A.12 illustrates the value of the last row of $F$ (i.e. the masking for the 512th token) for each of the layers. We observe some interesting patterns, for example, layer 4 persists the end-of-sentence periods ('.').

# 8 RELATED WORKS

**Transformer Improvements.** Our work aims at improving the transformer architecture. Since its introduction in Vaswani et al. (2017), a large volume of research proposed architecture modifications towards an improved model. Some notable works here include Pre-LN instead of Post-LN (Xiong et al., 2020), gated units instead of MLPs (Shazeer, 2020), removing the biases (Raffel et al., 2023), using RMSNorm instead of LayerNorm (Zhang & Sennrich, 2019), normalizing the Q and K projections (Dehghani et al., 2023), and multi-query and group-query attention (Shazeer, 2019; Ainslie et al., 2023).

**Attention Modifications.** Our work focuses on modifying the attention module. Most of the existing research work around modifying attention focuses on a different goal than ours, specifically on devising more *efficient* attention variants. Some such variants are based on approximations to the attention operations and include sparse attention approximations (Child et al., 2019; Ding et al., 2023), and linear attention approximations (Shen et al., 2024; Katharopoulos et al., 2020; Schlag et al., 2021). Some works focus on hardware-aware optimizations instead, such as FlashAttention (Dao et al., 2022) and Ring Attention (Liu et al., 2023a).

**Context Pruning.** A part of our work (Section 4) consists of removing elements from the context buffer. Several works employ this mechanism in order to increase inference efficiency. Among those are *compression* methods, that aim to learn a compressed representation for tokens in the context buffer, with or without auxiliary losses, in order to replace several tokens with their compressed form (Munkhdalai et al., 2024; Ren et al., 2023; Yun et al., 2023; Mu et al., 2024; Rae et al., 2019). Another line of work tries to simply remove elements from the context buffer without replacing them with new compressed forms, with minimal negative impact to model quality. The simplest and most widely adopted of these is using local attention windows in some of the layers (Wang et al., 2019; Beltagy et al., 2020; Zaheer et al., 2021). More sophisticated variants employ heuristics to evict less useful tokens from the context buffer instead of just the earliest ones (Oren et al., 2024; Zhang et al., 2023; Liu et al., 2023b; Berchansky et al., 2023; Ge et al., 2024). Anagnostidis et al. (2024) fine tunes existing models to learn to prune tokens from the context buffer with similarities to selective attention, but is more involved (e.g. needs root solving for evaluating the $\alpha$-sigmoid), produces binary prune decisions, and notably doesn't improve quality.

**Inference Efficiency.** A part of our work (Section 4) consists of making inference from transformers more efficient. Many approaches aim to speed up inference from transformers, including distillation (Hinton et al., 2015), sparsification (Jaszczur et al., 2021), quantization (Hubara et al., 2016), architecture modification (So et al., 2022; Shazeer, 2019), and algorithmic optimization (Dao et al., 2022; Leviathan et al., 2022).

Finally, we note that the importance of *learning to forget* has been shown repeatedly in many works, more generally beyond transformers. One of many notable examples are the forget-gates in LSTMs (Hochreiter & Schmidhuber, 1997).

# 9 DISCUSSION

In this work we introduced *Selective Attention*, a simple parameter-free change to the standard attention mechanism which consistently improves language modeling performance across model sizes and context lengths, and can lead to substantial inference efficiency improvements. Given that it adds no new parameters, only a negligible amount of compute, and provides consistent improvements, selective attention might be a good default for transformer decoders.

**Future directions.** We applied selective attention to decoder-only transformers. It could be interesting to investigate its applicability to encoders as well (see Appendix A.9 for initial results in an encoder-decoder setup); Reducing the size of the context as in Section 4 improves inference efficiency but not training efficiency. It might be interesting to explore iteratively reducing the size of the context buffer during training; We did not further train the models after removing elements as per Section 4. It seems conceivable that further improvements could be achieved with some additional training after context reduction; We only experimented with pre-training models with selective attention. It is interesting to investigate how it could be applied in a fine-tuning step to existing models; While we observed similar results with selective attention in several setups (Section 5), there are

still important variants we did not test, notably transformers with multi-query (Shazeer, 2019) and grouped-query (Ainslie et al., 2023) attention, as well as models much larger than 1B parameters; Finally, selective attention can be implemented in a GPU-aware way, similar to Flash Attention (Dao et al., 2022).

## 10   IMPROVING NEURAL ARCHITECTURES

In *The Art of Transformer Programming*, Leviathan (2022) selected a set of foundational problems (sorting, searching, addition, etc.) and manually implemented transformers to solve them (i.e. by manually setting the model's weights). They showed that several programs become much easier, especially for small transformers, when equipped with a mechanism allowing to selectively mask items in the context buffer, similar to selective attention. They further hypothesized that such a mechanism will have similar positive effects on language modeling, which motivated our work.

Zhou et al. (2023) proposed the *RASP-Generalization Conjecture*, that "Transformers tend to length generalize on a task if the task can be solved by a short RASP program which works for all input lengths", i.e. that problems that are easily solved by transformers are those that are easily solved by human programmers using RASP. It follows that problems that are not easily solved by humans using RASP are hard for transformers as well, and if we made those easier, by changing the transformer architecture (and respectively the capabilities of RASP) we could meaningfully improve transformers. Similarly, when constructing transformer programs by hand, Leviathan (2022) notes that ". . . the most interesting cases are those that are hard for us humans and are hard for the optimizer or the architecture, and understanding these better might be key to creating better AI systems."

We are strong advocates of this method, and believe that finding basic problems for which we cannot program a general solution by hand on a neural model, is a fertile approach for architecture improvements.

### ACKNOWLEDGMENTS

We'd like to extend a huge thank you to Raya Leviathan, Blake Hechtman, Asaf Aharoni, Uri Mendlovic, Danny Lumen, Avinatan Hassidim, Dani Valevski, Eyal Segalis, Molad Eyal, the Theta Labs and Google Research teams, and our families for insightful feedback, ideas, suggestions, and support.

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

# A APPENDIX

## A.1 TRANSFORMERS WITH SELECTIVE ATTENTION LEARN A GENERAL SOLUTION TO VARIABLE ASSIGNMENT

Transformers with selective attention reach close to 0 validation loss and 100% precision extremely fast when trained on `Variable Assignment` and they generalize well to out of distribution cases, unlike transformers without selective attention.

**Setup.** We train small transformers ($d = 3$), with and without selective attention, to solve the `Variable Assignment` problem with 3 variables, 1,000 possible values, and 128 assignments. We train with a batch size of 2,048 for 65,536 steps.

**In distribution.** The transformer with selective attention reaches a validation loss of 0.002 (and 100% accuracy) after less than 1,000 training steps. The transformer without selective attention only achieves a validation log-perplexity loss of 3.18 and 26% accuracy after 1,000 step. At the end of training (65,536 steps) the transformer with selective attention obtains a loss of 2.2e-8, whereas the transformer with standard attention is at 0.01. Both transformers reach 100% accuracy at the end of training.

**Out of distribution.** We observe much stronger generalization capabilities for the transformer with selective attention. When we run on an out of distribution test set with the same 3 variables but only 2 possible values, the transformer with standard attention's accuracy drops substantially to 70% (loss of 3.64). Meanwhile the transformer with selective attention maintains 100% accuracy (with a loss of 2.4e-8).

We observed similar results in other settings (e.g. 10 variables and 10 possible values). We also repeated the experiments with somewhat larger transformers ($d = 8$) and observed similar results.

See Appendix A.11 for an example of the attention patterns for the `Variable Assignment` task.

## A.2 ADDITIONAL SYNTHETIC TASKS

We validate selective attention on two additional synthetic tasks, `Copy` and `Parity`*, that are on the two opposite extremes in terms of memory requirements. See Figure 1.

**Setup.** We train small transformers ($d = 3$), with and without selective attention, to solve the `Copy` and `Parity`* tasks. We train with a batch size of 2,048 for 65,536 steps.

**`Copy`.** Here the transformer gets an arbitrary sequence of varying length delimited by a special token and is tasked with outputting a copy of the sequence. We used lengths that are uniformly distributed between 1 and 24. The context size for this task is $3 + 2 \times L_{max}$, where $L_{max} = 24$ is the length of the longest possible input sequence (the 3 extra tokens are for `<BOS>`, `<EOS>` and the special end-of-input-sequence token). To solve this task, the model cannot forget anything before copying starts, after which point it can mask out tokens that were already copied.

**`Parity`*.** Here the transformer gets a binary sequence where bits in the odd positions are random and bits in the even positions contain the parity of all bits in the earlier odd positions. Equivalently, bits in the even positions contain the XOR of the bits in the previous two positions, so the two previous positions are enough for computing the next token. The loss only considers the even positions.

Unsurprisingly, transformers with selective attention achieve practically 0 loss (and 100% accuracy) at the end of training, as do standard transformers without selective attention, on both tasks.

## A.3 SEPARATE BILINEAR FORM

We experiment with using a separate bilinear form instead of reusing the output of an existing attention head. We compare transformers (for $d = 8$ and $d = 12$) trained with selective attention on C4 for 524,288 steps, to similarly trained transformers where the selection function is implemented via a separate bilinear form (adding additional parameters and computation). The transformers with standard selective attention (i.e. sharing the outputs of an existing attention head) achieve the same or slightly better log-perplexities at the end of training (see Table 2).

Table 2: The log-perplexity on the validation set after 524,288 training steps for (1) standard attention, (2) selective attention with a separate bilinear form for the selection module (more parameters than the baseline), and (3) selective attention.

|  | $d = 8$ | $d = 12$ | Additional parameters |
|---|---|---|---|
| Standard attention | 2.96 | 2.68 | **No** |
| Selective attention (separate) | 2.91 | **2.63** | Yes |
| **Selective attention** | **2.90** | **2.63** | **No** |

## A.4 SELF-IMPACT

Table 3 compares the results of forbidding self-impact (i.e. not allowing a token to affect its own attention operation, as in Section 3.3) to those when allowing it (i.e. not shifting the matrix $S$). As can be seen, the shifting provides a small but consistent improvement.

Table 3: The average log-perplexity on the validation set of 3 training runs after 65,536 training steps for selective attention vs selective attention without shifting, for various model sizes.

|  | $d = 10$ | $d = 12$ | $d = 14$ | $d = 16$ | $d = 18$ | $d = 26$ |
|---|---|---|---|---|---|---|
| Selective attention (no shift) | 2.927 | 2.832 | 2.753 | 2.692 | 2.641 | 2.516 |
| **Selective attention** | **2.923** | **2.831** | **2.750** | **2.691** | **2.639** | **2.511** |

## A.5 ABLATING THE CONSTRAINTS

We ablate the 3 constraints selective attention applies to $S$ (see Section 3.2).

**Negative selection.** While with selective attention a token can decide to reduce attention to another token by all future attention operations, allowing a token to *strengthen* another token's contribution to all future attention operations does not make sense. Indeed, when removing this constraint (dropping the ReLU, so that $S$ can contain negative values) the training does not converge.

**Masking the `<BOS>` token.** Since several algorithms can benefit from the existence of the sentinel <BOS> token (Leviathan, 2022), it is plausible that masking it is detrimental. When we allow selective attention to mask the <BOS> token, we observe neutral to slightly worse results compared to standard selective attention where the <BOS> token is forced to never be selected, see Table 4.

**Self-masking.** Since selective attention reuses an existing attention head as the selection function, motivated by the absorption observation (see Section 3.1), it seems plausible that a token should never mask itself. Indeed, when we allow tokens to mask themselves (i.e. we stop zeroing out the diagonal of $S$) we observe worse results, see Table 5.

Table 4: The log-perplexity on the validation set after 524,288 training steps for (1) selective attention without the <BOS> constraint and (2) selective attention.

| $d$ | Selective Attention (w/o <BOS> constraint) | Selective Attention |
|---|---|---|
| 12 | 2.6409 | **2.6373** |
| 14 | 2.5486 | **2.5483** |
| 16 | 2.4750 | **2.4741** |
| 18 | **2.4153** | 2.4156 |
| 20 | 2.3674 | **2.3673** |
| 24 | 2.2909 | **2.2865** |

Table 5: The log-perplexity on the validation set after 524,288 training steps for (1) selective attention without the self-masking constraint and (2) selective attention.

| $d$ | Selective Attention (w/o self-masking constraint) | Selective Attention |
|---|---|---|
| 12 | 2.7348 | **2.7251** |
| 14 | 2.6510 | **2.6423** |
| 18 | 2.5261 | **2.5209** |

## A.6 PERPLEXITY-EFFICIENCY TRADE-OFF

Figure 6 illustrates the trade-off between perplexity gains and efficiency gains when pruning as in Section 4. See Section 6 for details.

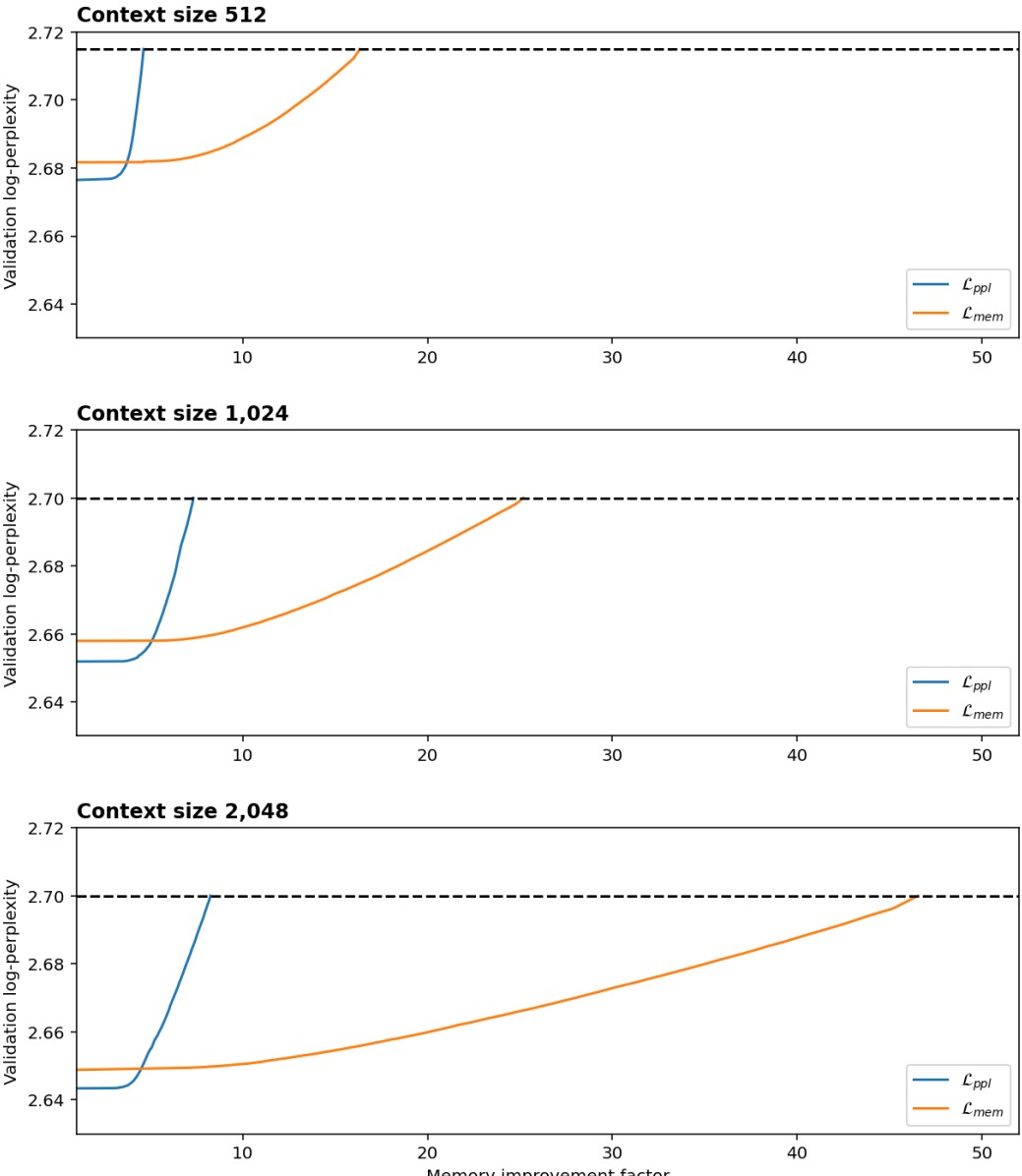

Figure 6: The trade-off between perplexity and KV-cache size for $d = 12$ transformers with context sizes of 512, 1,024, and 2,048. Note that in all cases the perplexity with selective attention is better or equal to that of the baseline without selective attention (the dotted lines). Selective attention transformers trained with the $\mathcal{L}_{mem}$ loss and $\epsilon = 0.1$ (Equation 2) match the perplexity of the baseline with 16X, 25X, and 47X less memory, while those with the standard loss match the perplexity of the baseline with 5X, 7X, and 8X less memory, respectively.

## A.7 EXAMPLE DETAILS

The following text from the C4 validation set was used in Figures 5, 11, and 12:

```
the real problem with traditional dental veneers has little to do
with how they function or their performance .  who determines what
a normal , aesthetically pleasing smile looks like is the real
issue .  america struggled for decades with defining " image "
as a marketplace bent on exploiting peoples ' flaws for economic
gain .  the danger of this national obsession has become systemic
since the days of twiggy .  fashion magazines offer photoshopped
perfection as the standard to which we should aspire .  the
effects of this insidious marketing made their way into breast
implants and the definition of a hollywood smile .  carving their
bodies and their teeth , people use their resources to chase a
false picture of their " perfect " self .  people make investments
in the tens of thousands of dollars at their dentist office to get
the " perfect " smile .  this message has become so endemic that
people with nice smiles are convinced only a " perfect " hollywood
smile is acceptable .  one particularly relevant example of this
is highlighted in a june 2015 article entitled " saving jane '
s smile . "  gary nankin , dds discusses how he " saved " the
smile of a patient who was not content with her first set of #
porcelain veneers .  1 .  endodontic referral for treatment of
tooth number 15 , followed by a composite core build - up .  2
.  periodontal therapy in both the anterior region and upper left
to achieve optimal tissue health .  4 .  preparation of maxillary
teeth and placement of permanent restorations .  5 .  placement
of dental implant by the periodontist followed by preparation of
mandibular teeth and placement of permanent restorations .  6 .
restore the now fully - healed and osseointegrated implant in the
position of tooth number 30 .  regarding a person ' s smile , the
strong link to self - esteem and self - worth make an imperfect
set of teeth a concern .  however , the picture in the article
clearly illustrates what appears to be a well - constructed and
healthy - looking smile .  the entire premise is puzzling .  how
does a dentist promote " saving " a smile that 97 % of the people
in america would love to show off ?  .
```

## A.8 COMPARISON WITH LOCAL ATTENTION

Table 6 compares the validation perplexity of $d = 12$ transformers with various local attention patterns (all-local and alternating), to that of a standard transformer and to that of a transformer with selective attention. For the all-local attention transformers we set all layers to be sliding window attention layers with a fixed sized window. For example, "All-local 32" denotes a transformer where all tokens can only attend up to 32 tokens back. We also include transformers with alternating local and global layers, where we have 3 local attention layers followed by 1 global attention layer, in a repeated fashion. For example, "Local-global 32" denotes a transformer with 3 local attention layers where tokens can only attend up to 32 tokens back, followed by a global layer where tokens can attend to all past tokens, and this 4-layer structure is repeated for the 12 layers of the $d = 12$ transformer. We report the perplexity numbers after 524,288 training steps. We observe that all local attention patterns perform worse than the dense baseline, which in turn performs worse than a transformer with selective attention.

Table 6: Validation log-perplexity of transformers with different local attention patterns.

| Model Type | Validation Log-Perplexity |
|---|---|
| All-local 32 | 2.7860 |
| All-local 64 | 2.7386 |
| All-local 128 | 2.7154 |
| All-local 256 | 2.6981 |
| All-local 384 | 2.6873 |
| All-local 448 | 2.6849 |
| All-local 480 | 2.6834 |
| | |
| Local-global 32 | 2.7046 |
| Local-global 64 | 2.7105 |
| Local-global 128 | 2.7154 |
| Local-global 256 | 2.6993 |
| Local-global 384 | 2.6895 |
| Local-global 448 | 2.6870 |
| Local-global 480 | 2.6861 |
| | |
| Baseline (standard attention) | 2.6815 |
| | |
| **Selective attention** | **2.6372** |

## A.9 RESULTS ON T5

We experimented with applying selective attention while pre-training T5 (Raffel et al., 2023). Here we use the standard T5 pre-training recipe and code from T5X (Roberts et al., 2022). Specifically, in this setup, the model is an encoder-decoder, and we apply selective attention to the decoder only, leaving the encoder as-is. The standard T5 span-corruption pre-training objective is used, as in Raffel et al. (2023), both for training and the reported metrics. See Table 7 for the results.

We observe some improvements for T5 with selective attention for the 3 tested model sizes.

Table 7: The span corruption loss per non-padding token on the validation set of 3 training runs after 524,288 training steps for a baseline T5 encoder-decoder vs a T5 encoder-decoder where the decoder is equipped with selective attention, for various model sizes.

| | T5-small | T5-base | T5-large |
|---|---|---|---|
| T5 | 1.962 | 1.693 | 1.528 |
| **T5 with selective attention** | **1.952** | **1.691** | **1.522** |

## A.10 COMPARISON WITH EFFICIENT ATTENTION METHODS

Figure 7 compares selective attention to other efficient attention and attention pruning methods, including $H_2O$ (Zhang et al., 2023), TOVA (Oren et al., 2024), and sparse attention (Child et al., 2019). We also include Window $+ 4$ following Oren et al. (2024). Note that $H_2O$, TOVA, and Window $+ 4$ can be applied post-training, whereas selective attention and sparse attention require training the model[3].

We observe that in the tested setting of language modeling on C4, selective attention substantially outperforms all of the tested efficient attention baselines.

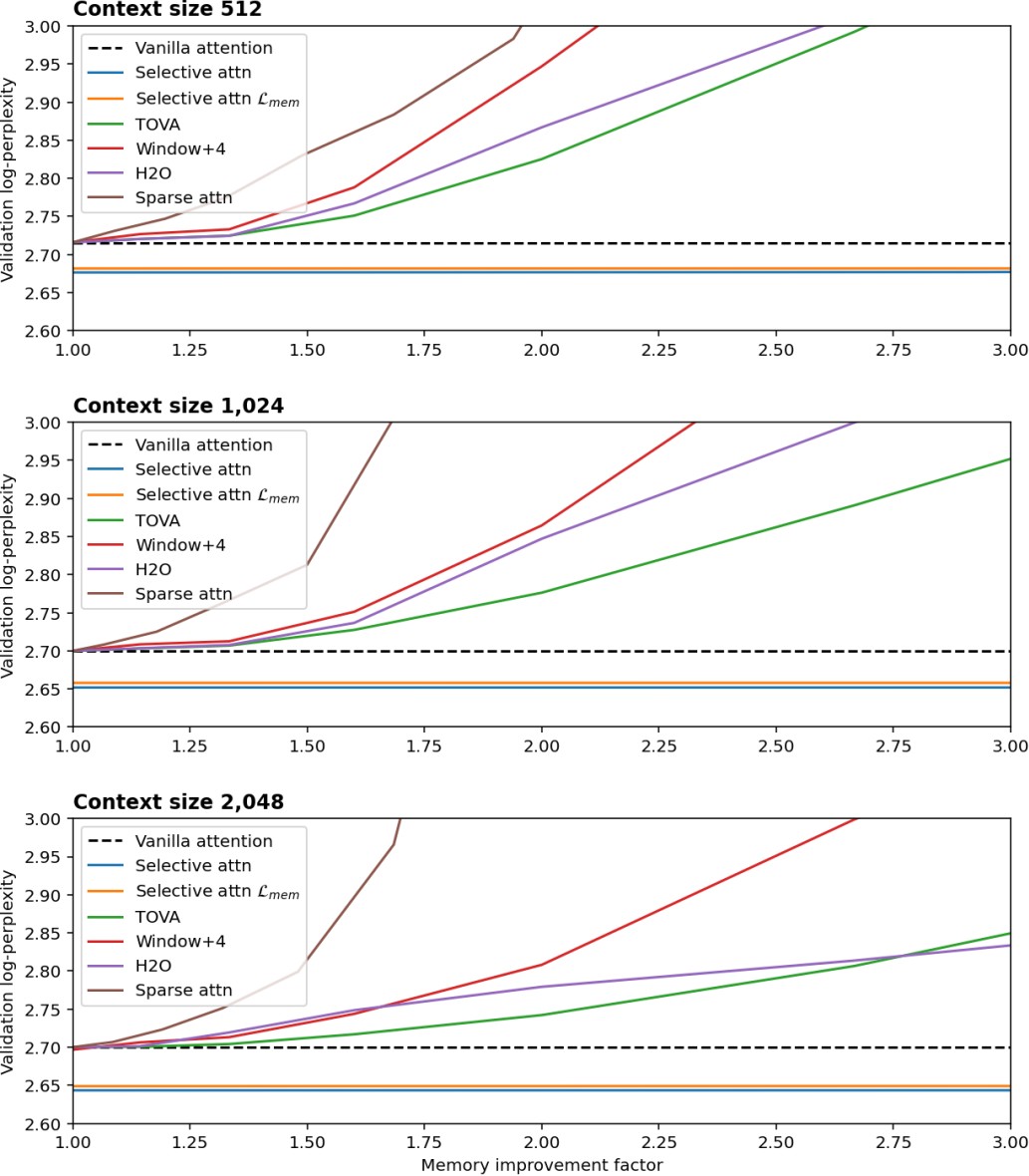

Figure 7: The trade-off between validation perplexity and KV-cache size for $d = 12$ transformers with context sizes of 512, 1,024, and 2,048 for selective attention and other efficient attention mechanisms.

---

[3]Also see Appendix A.8 for a comparison with models trained with various local attention patterns.

## A.11 EXAMPLE ATTENTION PATTERNS FOR VARIABLE ASSIGNMENT

Figure 8 shows an example of the attention patterns for all 3 layers of transformers with and without selective attention, for the Variable Assignment task. The part of the example sequence shown starts with the tokens: Z=, 177, Y=, 661, Z=, 114, Z=, 468.

For the first layer of the transformer with selective attention we observe that tokens of type "<value>" (e.g. 177, 661, etc., i.e., those at the even positions) attend to themselves and to the immediately preceding token (of type "<variable>=", e.g. Z=, Y=, etc.). A potential explanation is that this allows the value tokens to absorb the information of the variable they are assigned to, so that from this point onwards, the token would contain a representation of the pair (variable, value), and the preceding variable-only token would no longer be relevant for future tokens. E.g., the token 177 might contain a combined representation of Z and 177. In this same layer, tokens of type "<variable>=" attend to themselves.

We further observe that for the transformer with selective attention, the attention patterns in the second and third layers are almost identical. In these layers, tokens in the even positions, now containing a combined (variable, value) representation according to the postulate above, attend to themselves. Tokens in the odd positions, still containing a representation of the assigned variable according to the hypothesis above, attend to the value of the last assignment to the same variable. This allows masking the previous assignment (see Figure 1).

In contrast, the baseline transformer without selective attention (which doesn't solve the Variable Assignment task for out of distribution problems, see Appendix A.1), exhibits attention patterns that are harder to interpret.

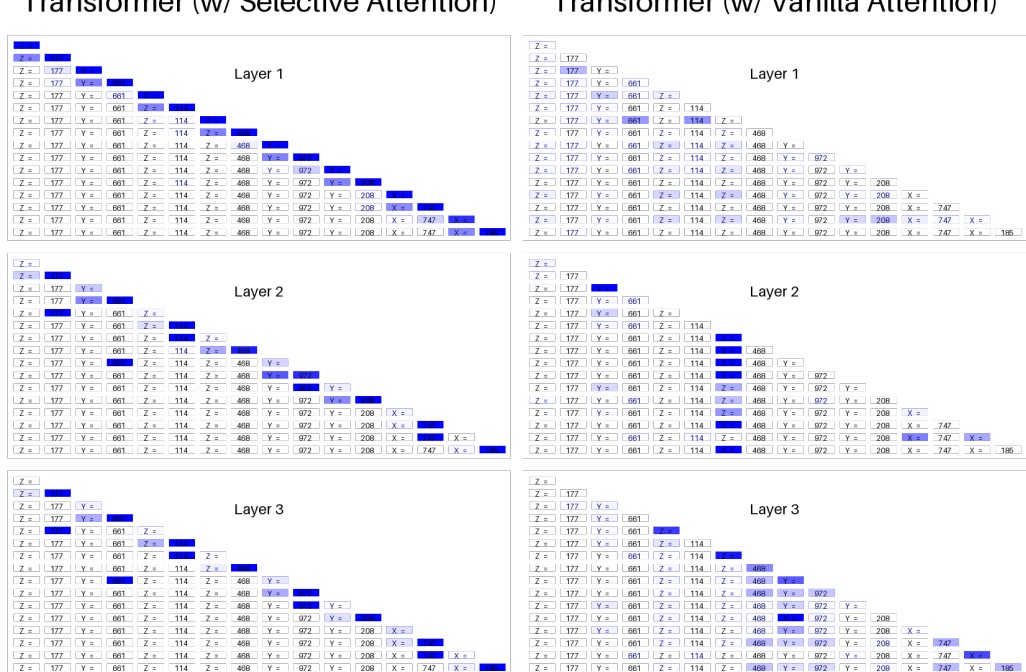

Figure 8: The attention patterns (attention strength averaged across heads) for all layers of transformers with selective attention (left) and with vanilla attention (right).

### A.12 ADDITIONAL FIGURES FOR CONTEXT PRUNING

Figure 9 illustrates the values of the $F$ matrix averaged across 1,000 examples of length at least 512 from the C4 dataset.

Figure 10 compares the values from Figure 9 to those obtained from a different training run (different random initialization and different data shuffle). While this isn't always the case, we sometimes observe stable sparsity patterns like those in this example, hinting at some general properties of language modeling on C4.

Figure 11 illustrates the tokens that remain in the context buffer after pruning (as in Section 4) for the example text (see Appendix A.7), for a $d = 12$ model with selective attention, trained with the $\mathcal{L}_{mem}$ loss (Equation 2, $\epsilon = 0.1$), for a memory budget where the validation perplexity matches a transformer without selective attention. The per-layer memory budgets chosen by the pruning algorithm for this model are: `[8, 48, 8, 8, 24, 8, 168, 16, 8, 64, 8, 8]`, leading to a memory saving factor of 16X.

Figure 12 shows which tokens are pruned for the example text in Appendix A.7.

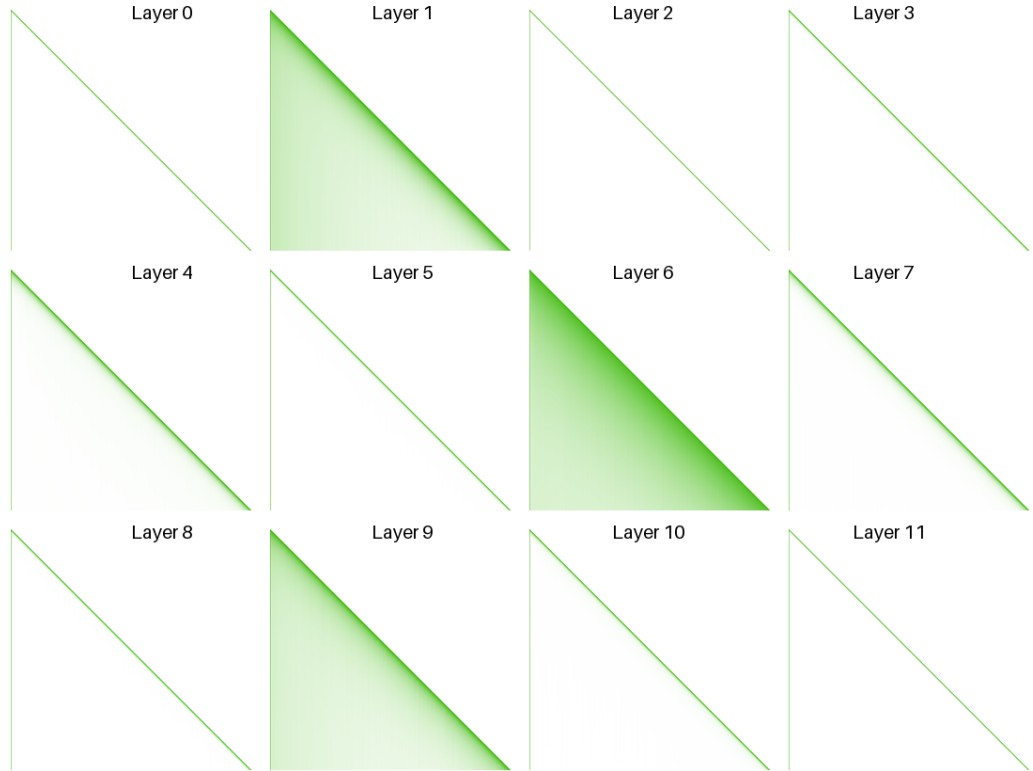

Figure 9: Visualization of the $F$ matrix (greener is lower, i.e. less masking) for a $d = 12$ transformer averaged across 1,000 examples.

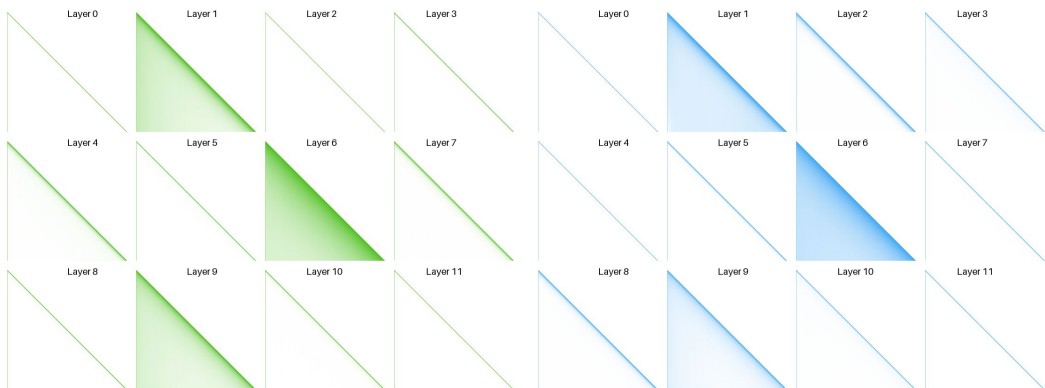

Figure 10: Visualization of the $F$ matrix (greener/bluer is lower, i.e. less masking) for a $d = 12$ transformer averaged across 1,000 examples for two training runs (different random initialization, and different shuffle of the training data). While we only have anecdotal evidence, it is interesting that we sometimes observe these stable sparsity patterns across training runs.

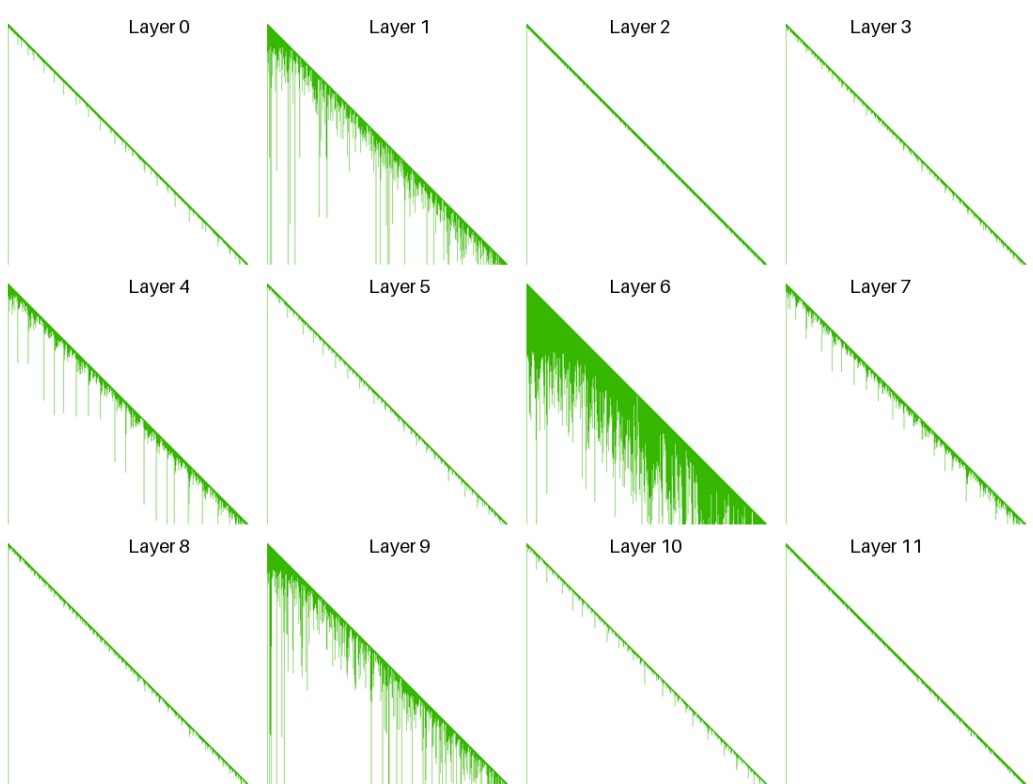

Figure 11: Visualization of the persisted elements for a $d = 12$ transformer for the text in Appendix A.7. The white pixels denote tokens removed from the context buffer as per Section 4.

Figure 12: A visualizing of the elements that are masked for the last (512th) token, for a $d = 12$ transformer for the text in Appendix A.7. We observe some interesting patterns, for example, layer 4 persists the end-of-sentence periods (".").

## A.13 PARAMETER COUNTS

Table 8 shows the actual parameter counts for models with different $d$s (see Section 5). Note that selective attention does not add any extra parameters.

Table 8: The number of model parameters for different values of $d$ as in Section 5.

| $d$ | $n_{layers}$ | $n_{heads}$ | $d_{model}$ | Number of parameters |
|---|---|---|---|---|
| 8 | 8 | 8 | 512 | 33,603,584 |
| 10 | 10 | 10 | 640 | 59,699,200 |
| 12 | 12 | 12 | 768 | 97,615,872 |
| 14 | 14 | 14 | 896 | 149,666,048 |
| 16 | 16 | 16 | 1024 | 218,226,688 |
| 18 | 18 | 18 | 1152 | 305,717,760 |
| 20 | 20 | 20 | 1280 | 414,387,200 |
| 22 | 22 | 22 | 1408 | 546,641,920 |
| 24 | 24 | 24 | 1536 | 704,950,272 |
| 26 | 26 | 26 | 1664 | 891,468,032 |
| 28 | 28 | 28 | 1792 | 1,108,645,888 |

