# OpenReview forum: "Selective Attention Improves Transformer"
_ICLR.cc/2025/Conference — ICLR 2025 Poster_

### Official Review · Reviewer_obGb · 2024-10-17

**Soundness:** 3
**Presentation:** 3
**Contribution:** 4
**Rating:** 8
**Confidence:** 4

**Summary:**

This paper presents selective attention, a method for adjusting the attention weight of tokens based on the amount of attention they received from previous tokens. The authors also propose to use this method to drop some of the history tokens with the lowest (adjusted) weight, also proposing a new loss term to transformer-based LLM training. Extensive experiments show that this method both consistently improves performance, and reduces memory requirements substantially.

**Strengths:**

- The method is simple and intuitive
- Results are very strong and consistent, showing both accuracy improvements, and very impressive memory reduction
- Experiments are quite extensive
- Given these results, overall I tend to agree with the authors' statement in Section 9: " Given that it adds no new parameters, only a negligible amount of compute, and provides consistent improvements, *selective attention might be a good default for transformer decoders*."

**Weaknesses:**

While I like this paper a lot, I have several reservations.

First, the level of novelty is not huge. The idea of pruning the least important token in each iteration is similar in spirit to cache-pruning methods such as H20 (Zhang et al., 2023), TOVA (Oren et al., 2024) and others. The authors basically apply a similar approach at training time. While this method works great (as noted above), which is a sufficient contribution on its own, and also present some novel concepts (e.g., the new loss term), it would be better to tone down the novelty claims a bit.

Second, I am not sure I am fully convinced by some of the major claims in the paper.

(a) the examples in Section 2 are very appealing, but they are missing a critical baseline: how much attention does the vanilla transformer assign the pruned tokens in such cases? Perhaps it already knows it needs to ignore them? Again, the method works great so obviously something good is happening, but it is not clear the intuition is capturing the essence of it.

(b) similarly, the authors say (#113) "if token b has determined that token a is irrelevant or even misleading to future tokens such as c, there is nothing it can do in the given layer to correct for this.". But I am not sure how their method allows the transfer of such negative signal. If I understand correctly, the authors penalize tokens that received high attention in previous steps (I.e., their value in F is high). I am not sure how information about irrelevant or misleading tokens can be propagated in such cases.

Third, while the writing is generally clear, I found section 3 a bit confusing, requiring several reads to understand a fairly simple method. For instance, it is never explicitly mentioned that each token is penalized by the amount of attention it got from all subsequent tokens. I would also advise the authors to formally define masking, as it is not entirely trivial in this context.

**Questions:**

The authors mention that they compute the S matrix using "one of the existing heads" (#167). Their code indicates that they simple take the first one. Is this ablated in any way? Does taking a different head / an average of all heads make much difference?

---

> ### Author Response · Authors · 2024-11-21
>
> Thank you for your thorough review and great suggestions!
>
> Regarding your comments:
>
> **(a)** Thanks for pointing this out. We generated attention plots for variable assignment for a baseline transformer and, as expected, we see that attention is paid to irrelevant assignments (it's hard to draw strong conclusions from this beyond the empirical results, especially since several layers are involved).
>
> **(b)** The propagation of the information stems from the cumsum operation. Specifically, when a token “c” attends to a past token “a”, we penalize this attention weight by the total amount of attention the tokens between “a” and “c” paid to “a” (with the selection head). In this way, token “b”, between “a” and “c”, is able to make future tokens “c” not pay attention to token “a” simply by strongly attending to it with the selection head. For example, in the case where token “a” is “Bar” (which might have a misleading meaning of a physical bar), the next token “ack” might pay a large amount of attention to the previous token “Bar” with the selection head, therefore preventing future tokens from attending to “bar”.
>
> **(c)** Thank you - really great point! We added a couple of sentences to the method section following your suggestion and we think it made it much easier to understand!
>
> **(question)** note that without selective attention, all heads in a transformer are symmetrical, so there is no difference between using head 0 vs any of the other heads (the transformer with selective attention makes whichever head is used for selection special, breaking symmetry during training). It’s a really interesting suggestion to try an average of all heads - we did not try it and hope to do so in future work. We did experiment with other regimes that did not perform as well (e.g. a separate bilinear form as we show in appendix A3). We hope to pursue further variants in future work.

---

> > ### Comment · Reviewer_obGb · 2024-11-21
> >
> > Thank you for your response.
> >
> > (a) can you point to these plots? I am not sure where to find them in the paper.
> >
> > (b) I agree with your motivation, but it still doesn't capture the case I was talking about. If token b considers token a as irrelevant, and thus does not attend to it at all, then by definition this information cannot be propagated to c, right?
> >
> > (c) thanks! can you point out the specific changes?
> >
> > Also note about my first comment about novelty that was not addressed in your response.

---

> > > ### Author Response · Authors · 2024-11-22
> > >
> > > * (a) Sorry about this - the diagrams (with some explanations) are in the supplementary material now (“varass_attention_patterns.pdf”). Please tell us if in your opinion it's important that we add them to the appendix of the paper itself.
> > >
> > > * (b) Let’s consider 2 cases (1) where token “b” thinks token “a” is relevant for itself but irrelevant for future tokens, and (2) where token “b” thinks token “a” is irrelevant for all tokens (itself included):
> > >
> > >     - (1) Token b thinks token a is relevant *for itself* (i.e. it wants to read information from it) but token b also thinks token a is irrelevant *for other tokens* (and wants to hide token a from them). This case, where token b “absorbs” token a, is an important motivation for the work. In this case, token b attends to token a with the selection head, reads its meaning and hides it from the following tokens. An example here are the tokens “Bar” and “#ack”.The token "#ack" attends to "Bar", absorbs its meaning (so it now contains a representation for “Barack”), and from this point on the token “Bar”, which is irrelevant for future tokens, is hidden from them.
> > >
> > >     - (2) Token b knows that token a is irrelevant both *for itself and for future tokens*, and decides to attend to it without absorbing its meaning, but only to mask it from future tokens. This is the behavior we might see with the variable assignment task, where an assignment to a variable attends to previous assignments of the same variable even though it doesn't need any information from these tokens, but only in order to hide these misleading tokens from future attention operations.
> > >
> > > We hope this makes things clearer - please let us know otherwise!
> > >
> > > * (c) The main change is at the beginning of section 3, where we added the following: "Selective attention is a simple modification on top of standard attention. **The key idea is that tokens can mask previous tokens, i.e. the amount of attention a token c pays a previous token a can be reduced by the tokens located between a and c.**". We also changed the language in the paper to use the term “mask” more consistently (e.g. we previously used the terms “mask” and “soft-mask” interchangeably).
> > >
> > > * (novelty) Pruning less important tokens to improve inference efficiency has been previously explored in other great prior works like H2O and TOVA (as we discuss in the "Related Works" section). We definitely don’t claim any novelty here - please let us know if you think there is any language which might imply otherwise. We do think there are several important novel ideas in the paper which we did try to highlight, most importantly that enabling tokens to mask out other tokens during training, leads to *substantially improved generation quality and allows transformers to solve problems they previously couldn't*, such as Variable Assignment. The efficiency gains during inference emerged as a natural side benefit once we had a signal indicating which tokens the model decides to mask out. To the best of our knowledge, prior methods have not demonstrated that such masking improves model quality; instead, they often do the opposite - trading off some quality for efficiency.

---

> > > > ### Comment · Reviewer_obGb · 2024-11-26
> > > >
> > > > Thanks for the clarifications.
> > > >
> > > > (a) thanks. I would consider putting it in the main paper (at least the findings, the figure can go in the appendix). I think it strengthens your claims
> > > > (b) what about case 3? the model finds the token irrelevant and decides **not** to attend to it? as a result, future tokens will not get any negative signal

---

> > > > > ### Author Response · Authors · 2024-11-26
> > > > >
> > > > > **(a)** Thanks. We cleaned up the figure and text and added them to a new Appendix (A11). For the main text, we prefer to focus on the empirical results (that variable assignment is solved in the general case only with selective attention) as the example is illuminating, but anecdotal. Exploring the attention patterns with selective attention more rigorously and for various tasks sounds like a really interesting direction for future work.
> > > > >
> > > > > **(b)** Correct, there could be cases where the model would choose to not propagate information (e.g. if like you say, token b chooses not to attend to token a with the selection head, or more generally, token b could still attend to token a with the selection head, but using a negative attention logit, which would not propagate information due to the relu we apply on the selection head). Selective attention is a mechanism that allows information to propagate when it is beneficial (and as is evident from the results, the model seems to learn to use this mechanism well).

---

### Official Review · Reviewer_T9JB · 2024-11-04

**Soundness:** 3
**Presentation:** 3
**Contribution:** 3
**Rating:** 5
**Confidence:** 4

**Summary:**

This paper argues that, for transformer-based language models, selectively attending to a subset of the tokens in the context can improve the model's quality. Motivated by this intuition, the paper proposes a new attention technique, where if a token has already received high attention weights by previous tokens, it means that its content has already been "absorbed" in the model's representations, and therefore it should receive less attention weights onward. The paper proposes an implementation for this idea by reusing the attention logits of one of the attention heads, and thereby adding no additional parameters and only a small amount of computational overhead. Further, the proposed method also allows for pruning some tokens from the context when they will never be attended to again, which leads to memory saving.

Experiments with a 100M transformer based LM trained on the C4 dataset suggests that the proposed approach achieves promising results on language modeling perplexity and one downstream task (HellaSwag).

**Strengths:**

- The motivation is clear, and the execution of the idea to not to tokens that are already attended to is clever and clean
- The proposed method is easy to implement, adds no parameters, and saves memory overhead
- Interesting results and analysis

**Weaknesses:**

- My main concern is about the limited empirical evaluation where only one downstream task is considered. As the paper argues, "different tasks have different requirements," it is crucial to explore whether selective attention is broadly applicable by evaluating it on a diverse set of tasks with different requirements. This can be complemented with synthetic evaluations that might require the model to store the entire sequence, e.g., counting the number of a certain token in a sequence.
- Besides a more comprehensive evaluation, I also encourage the authors to strengthen their findings by trying selective attention on larger-scale models and datasets

**Questions:**

None

---

> ### Author Response · Authors · 2024-11-21
>
> Thank you for your great suggestions, especially regarding empirical evaluation.
>
> To address your comments, we:
> 1. Evaluated selective attention on 5 additional downstream tasks (in addition to HellaSwag) and observed clear improvements with selective attention in all of them. We updated table 1 in the main text to include the results for all 6 tasks (see updated pdf).
> 2. Tested selective attention on 3 synthetic tasks with different memory requirements, including  the extreme example where storing the entire sequence is needed - for copying an arbitrary string. We added some details to Appendix A1/A2 in the new pdf (transformers with selective attention solve all of these, including copying, perfectly).
>
> Regarding the suggestion to try selective attention on larger-scale models - you might have missed this, since your summary says that we only tested models with 100M parameters, but we did train models of varying sizes from 30M to 1.1B. For example, a d=28 model has 1.1B parameters (see Appendix A13 in the new pdf). We include results for all of these model sizes throughout the paper (for example, see Figure 3 right, or Table 1). We do not have resources for training models much large than ~1B parameters, but we did observe that the gains persist across all tested model sizes.
>
> Hopefully the above addresses your concerns!

---

> > ### Comment · Reviewer_T9JB · 2024-11-27
> >
> > Thanks for clarifying my misunderstanding and addressing some of my concerns. I have raised my score.

---

### Official Review · Reviewer_5FJ3 · 2024-11-04

**Soundness:** 3
**Presentation:** 1
**Contribution:** 3
**Rating:** 6
**Confidence:** 3

**Summary:**

This paper introduces *Selective Attention*, a parameter-free modification to the standard attention mechanism in transformers which helps reduce the attention given to irrelevant elements in the context. To do so, they introduce a *selection mask* in the attention computation to mask “irrelevant” token, and propose using the previous tokens’ attention on other tokens (on a specific head) as masking for future attention computations. On language modelling on C4, results show that transformers equipped with Selective Attention can achieve comparable performance to standard transformers with twice the attention heads and parameters, and slightly better performance on the downstream task of HellaSwag.

The authors then propose a method that uses the selection mask to prune elements from the attention's context buffer, reducing memory and computational demands during inference, and show that it can lead to significant memory saving (up to 47x for large context sizes) while preserving performance of non-selective attention transformers.

**Strengths:**

- Given the simplicity of the proposed approach and the reported performance gains, it seems that selective attention could be a significant addition to the transformer architecture if it is further validated.
- Even without the performance gains, the efficiency gains (particularly without having to modify the pretraining loss) are quite relevant and make Selective Attention look like a viable alternative to other efficient attention mechanisms.

**Weaknesses:**

- The main problem of this paper is weak experimental validation. The authors only show gains in the language modelling task (using the relatively noisy and deprecated C4 dataset) and on a single downstream task (where they show smaller gains). They don't compare to existing pretrained models or other efficient attention mechanisms. While the (limited) experimental results are promising, they are not enough to validate the approach. I suggest replicating the recipe of an existing (state-of-the-art) pretrained LLM, replacing the attention mechanism with Selective Attention. Additionally, in the current LLM area, more downstream tasks need to be tested to validate the approach. For the efficiency section, the authors should compare with other efficient attention mechanisms.
- The presentation paper could also be better: it uses a whole page for a single figure and further page for intuition, both of which could be much shorter. It then moves decently important plots to the appendix. I suggest restructure the paper to give more space to the experimental results and comparison to other methods.

**Questions:**

- In the selection attention computation, why use a single head’s logits to mask all heads? Did you try keeping each head separate (i.e. using the logits head i for the selection mask of head i)?

---

> ### Author Response · Authors · 2024-11-21
>
> Thank you for your great suggestions, especially on improving the experimental validation.
>
> Following your comment, we:
> 1. Evaluated selective attention on 5 additional downstream tasks (in addition to HellaSwag) and observed clear improvements with selective attention in all of them. We updated table 1 in the main text to include the results for all 6 tasks (see updated pdf).
> 2. Following your suggestion to follow the recipe of an existing pre-trained model, we pre-trained a standard T5 encoder-decoder, as well as one where we apply selective attention to the decoder. Here we used the standard T5 span-corruption pre-training task (not language modeling). We used the existing T5X codebase for the baseline, and the same codebase with selective attention to make sure the standard recipe is indeed followed. We observe gains in this setup as well - see appendix A9 in the updated pdf.
> 3. We added comparisons to popular efficient attention methods such as H2O, TOVA, and sparse attention (in addition to sliding window attention and larger attention modules) as suggested. See Appendix A10 in the revised pdf (the models with sparse attention are still training - we will update again when the final figures are ready in a couple of days). Selective attention achieves a substantially better quality-cost tradeoff than all tested baselines.
>
> Hopefully the above addresses your concern regarding experimental validation (we think this really improved the paper, so thank you again for highlighting the issue and the detailed suggestions for how to address).
>
> Regarding your question: that’s a great point. We tried this and it did not work as well in our experiments. We do think there is much more to explore in this direction with other similar modifications.

---

> > ### Comment · Reviewer_5FJ3 · 2024-11-22
> > **Response to Rebuttal**
> >
> > Thanks for the response. The new experiments do make the validation of the approach, but I still think the presentation of the paper is non-standard (i.e. the new T5 and sparse attention results should go in the main paper rather appendix)
> > Nevertheless, I decided to raise my score.

---

### Official Review · Reviewer_L9tT · 2024-11-04

**Soundness:** 3
**Presentation:** 3
**Contribution:** 3
**Rating:** 8
**Confidence:** 3

**Summary:**

The paper introduces a parameter-free modification to the transformer attention mechanism called Selective Attention, which reduces the attention to irrelevant or unneeded tokens in a sequence. This approach enhances performance in language modeling tasks by reducing memory usage and computational costs without compromising model quality. Experiments show that transformers with Selective Attention achieve comparable performance to larger models with traditional attention mechanisms, providing improved efficiency and effectiveness across various model sizes and tasks.

**Strengths:**

1. **Efficient Memory Management**: The Selective Attention mechanism effectively prunes unneeded tokens, significantly reducing memory usage during inference without degrading model performance. This efficiency gain is particularly valuable for scaling transformers in resource-constrained environments.
2. **No Additional Parameters**: Selective Attention operates without introducing new parameters or significantly increasing computational overhead, which preserves the simplicity of the transformer architecture.
3. **Theoretical Motivation and Experimental Rigor**: The paper provides a thorough theoretical motivation for selective attention, supporting it with extensive experimental results on benchmarks like HellaSwag and C4, which demonstrate the effectiveness of the approach.

**Weaknesses:**

1. **Limited Scope of Model Architectures**: The experiments are primarily conducted on decoder-only transformer models. Further analysis is needed to verify if Selective Attention can similarly benefit encoder models or encoder-decoder models used in other tasks, such as translation or summarization.
2. **Potential Over-Reliance on Hyperparameter Tuning**: Selective Attention’s performance may depend on optimal memory budget settings per layer, which could complicate deployment in different tasks or models. Although a memory reduction process is described, further tuning could be required in practical applications.
3. **Dataset and Task Diversity**: While Selective Attention shows improvements in language modeling tasks, testing it on a wider range of tasks (e.g., text generation or long document understanding) would strengthen the case for its generalizability and adaptability to diverse applications.
4. **Comparison with Similar Methods**: The paper could benefit from a more detailed comparison with other recent efficient attention mechanisms, such as sparse attention or adaptive attention approaches, to highlight any relative advantages Selective Attention may have.

**Questions:**

1. Have you considered evaluating Selective Attention on other architectures, such as encoder or encoder-decoder transformers? If so, could you share any preliminary results or insights?

2. How sensitive is the performance of Selective Attention to the memory budget per layer? Could you offer guidelines on tuning this budget for tasks beyond language modeling?

3. Have you tested Selective Attention on tasks other than language modeling, such as question answering or text summarization? How might the approach handle tasks with varied context and attention needs?

4. Could you provide further comparisons between Selective Attention and other efficient attention methods, such as sparse or linear attention, to clarify the unique advantages of Selective Attention?

5. How does context pruning with Selective Attention affect tasks requiring long-range dependencies? Are certain types of tokens or information more prone to being pruned?

---

> ### Author Response · Authors · 2024-11-21
>
> Thank you for the time and effort you put into reviewing our paper. Hopefully these answers and updated pdf address your questions and concerns.
>
> **Application to encoder/encoder-decoders:** this is a great point. It is not clear to us how to apply selective attention in the encoder-only case (e.g. the cumsum might not make sense). Based on your comment we experimented with adding selective attention to a T5 encoder-decoder by keeping the encoder as-is and applying selective attention to the decoder as in the paper (see Appendix A9 in the updated pdf). We observed gains here as well.
>
> **Memory-budget tuning:** Note that the quality gains are obtained without any tuning. For also achieving the memory efficiency gains via pruning, tuning the memory budget to a suitable trade-off between efficiency and quality (as shown in Figure 6) is a simple process.
>
> **Task diversity:** Great point - thank you. Following this comment we ran experiments on 5 additional downstream tasks (in addition to HellaSwag) and observed clear improvements with selective attention in all of them. We amended Table 1 in the main text to include the results for all 6 tasks. Also, the T5 encoder-decoder experiments, which we mentioned above, are trained with the standard T5 span-corruption pre-training task (not language modeling) - see appendix A9 in the updated pdf. Finally, note that we also tested selective attention on 2 tasks on both extremes of memory requirements (copy and parity*) - following your comment, we added details (both selective attention and the baseline solve these perfectly) see Appendix A2.
>
> **Comparison to Other Attention Variants:** This is a great point. We added comparisons to popular efficient attention methods such as H2O, TOVA, and sparse attention (in addition to sliding window attention and larger attention modules) as suggested. See Appendix A10 in the revised pdf (the models with sparse attention are still training - we will update with the final figure in a couple of days). Selective attention achieves a substantially better quality-cost tradeoff than all tested baselines.
>
> **Answer to questions:**
> 1. See the answer re encoder-decoder above and Appendix A9 in the updated PDF.
> 2. Good question. Note, as above, that the quality gains are available without any memory budget tuning. Figure 6 in the appendix shows the sensitivity of the savings in the case of pruning. For example, in the case of a context size of 2,048 with the explicit objective, we get up to about 10X reduction in memory requirements without any effect on quality, and then up to 47X (when we match the quality of the baseline without selective attention). For non-language modeling tasks, we recommend following the same procedure as in Section 4, we amended the phrasing to make this clearer in the new pdf.
> 3. Hopefully this was answered above (see Task Diversity).
> 4. Hopefully this was answered above (see Comparison to Other Attention Variants).
> 5. Great question. We observe great performance on the synthetic copy task which requires long range dependencies (we added more details in new Appendix A2). Also, in the synthetic task of variable assignment, we observe that a transformer with selective attention finds a *general* solution (unlike the baseline), so e.g. even when testing out-of-distribution cases where we assign a value to a variable at the beginning and don’t overwrite it for a long context until we query about it (requiring information from the beginning of the sequence), a transformer with selective attention manages to fetch that value easily. To summarize, we have 3 examples of selective attention working well with long-range dependencies: copy, variable assignment, and language modeling.
> Regarding which tokens are more likely to be pruned, this is a super interesting question. We gathered a lot of interesting anecdotal evidence (a couple of anecdotal notes are in the paper, e.g. that one of the layers of a transformer trained on C4 prunes everything except the end of sentence periods). Other interesting findings (that we did not include in the paper) are that some layers save specific parts of speech, we hope to explore this more fully in future work.

---

> > ### Comment · Reviewer_L9tT · 2024-11-27
> > **Response to rebuttal**
> >
> > Thank you for your response. The new experiments validate the approach further, so I have decided to raise my score.

---

### Author Response · Authors · 2024-11-21

Thank you to all reviewers for the thorough reviews, recognizing the impact, simplicity and strengths of our method, and the great comments. Based on your comments we have made several important changes that we think really improve the paper - thank you!

Please see individual answers for more details. Here is a summary of the main additions:

1. Added an evaluation of a baseline transformer vs a transformer with selective attention on 6 downstream tasks (updated Table 1).
2. Added an evaluation of selective attention on the standard T5 encoder-decoder and the T5 span-corruption objective (Appendix A9).
3. Added a comparison with other efficient attention baselines, including H2O, TOVA, and sparse attention*, in addition to sliding window attention (Appendix A10).

We observe gains with selective attention in all of the above.

Please see more details and answers in the individual replies.

We hope that these address your concerns and that you consider raising our scores.

Thank you again,

The authors

\* Note that the models with sparse attention are still training and we will update here and the pdf with the final figures when training finishes in a couple of days.

---

> ### Author Response · Authors · 2024-11-24
>
> Quick update - the models with sparse attention finished training and we uploaded a new pdf with all promised changes (the sparse attention graphs are in Figure 7). Thank you!

---

### Meta-Review · Area_Chair_P8Rx · 2024-12-23

**Metareview:**

### Summary
This paper presents an elegant method for selective attention that prune tokens already absorbed in previous representations. Thus the pruned tokens can be evicted from the kv-cache to improve compute and memory usage.

### Strengths
The presented method is intuitive and simple. Results are very strong and consistent, showing both accuracy improvements and memory reduction. Visualizations of motivating examples are easy to understand.

### Weaknesses
The description of the method is not super clear, and it takes more than pass to fully understand it.

**Additional Comments On Reviewer Discussion:**

During the rebuttal period, reviewers raised the following key points:

1. Presentation is not clear and complete, e.g. lack of visualization of full attention examples. Authors revised their paper to make the method description more clear and also added visualization examples.

2. Reviewers complained that experiments are not sufficient, but authors added new experiments on more tasks and datasets and again got consistent improvements.

---

### Decision · Program_Chairs · 2025-01-22

Accept (Poster)